# Discovering state equivalences in UCT search trees by action pruning

## Abstract

A core challenge of Monte Carlo Tree Search (MCTS) is its sample efficiency which can be addressed by building and using state and/or action abstractions in parallel to the tree search such that information can be shared among nodes of the same layer. Though action abstractions are mostly easy to find in algorithms such as On the Go Abstractions in Upper Confidence bounds applied to Trees (OGA-UCT), nearly no state abstractions are found in either noisy or large action space settings due to constraining conditions. We provide theoretical and empirical evidence for this claim, and we slightly alleviate this state abstraction problem by proposing a weaker state abstraction condition that trades a minor loss in accuracy for finding many more abstractions. We name this technique Ideal Pruning Abstractions in UCT (IPA-UCT), which outperforms OGA-UCT (and any of its derivatives) across a large range of test domains and iteration budgets as experimentally validated. IPA-UCT uses a different abstraction framework from Abstraction of State-Action Pairs (ASAP) which is the one used by OGA-UCT, which we name IPA. We generalize both IPA and ASAP by introducing p-ASASAP and ASASAP abstractions.

## 1 Introduction

A plethora of important problems can be viewed as sequential decision-making tasks such as autonomous driving (Liu et al., 2021), energy grid optimization (Sogabe et al., 2018), financial portfolio management (Birge, 2007), or playing video games (Silver et al., 2016). Though arguably state-of-the-art on such decision-making tasks is achieved using machine learning (ML) as demonstrated by DeepMind with their AlphaGo agent for Go (Silver et al., 2016) or OpenAI Five for Dota 2 (Berner et al., 2019), there is still a demand for general domain-knowledge independent, on-the-go-applicable planning methods, properties which ML-based approaches usually lack but which are satisfied by Monte Carlo Tree Search (MCTS) (Browne et al., 2012) , the method of interest for this paper. For example, Game Studios rarely implement ML agents as they have to be costly retrained whenever the game and its rules and updated. Though not within the scope of this paper, improvements to MCTS might also potentially translate to ML-based methods that use MCTS as their underlying search.

One research area to improve MCTS is using abstractions that aim at reducing the search space by grouping states and actions in the current MCTS search tree to enable an intra-layer information flow (Jiang et al., 2014; Anand et al., 2015; 2016) by averaging the visits and returns of all abstract action nodes in the same abstract node used for the Upper Confidence Bounds (UCB) formula in the tree policy which increases the sample efficiency. However, one key weakness of state-of-the-art abstraction algorithms such as On the Go Abstractions in Upper Confidence bounds applied to Trees (OGA-UCT) (Anand et al., 2016) is that they struggle to find meaningful state abstractions given a reasonable computational budget even when the environment has a moderate action space size and stochastic branching factor as will be later illustrated in Section 3. Hence, they are essentially action abstractions for 1-step Markov Decision Processes (MDP) (Sutton & Barto, 2018) that are applied layerwise.

In this paper, we tackle exactly this problem by proposing a novel algorithm that directly aims at finding correct state abstractions that are not detected by Abstraction of State-Action Pairs in UCT

(ASAP-UCT) (Anand et al., 2015) or OGA-UCT to enable the detection of more Q node abstractions to ultimately boost the performance. The contributions of this paper can be summarized as follows:

**1.** Based on a theoretical justification and an empirical analysis we demonstrate the serious drawback of current SOTA approaches of finding sufficient state abstractions.

**2.** We propose **I**deal **P**runing **A**bstractions in **UCT** (IPA-UCT), an OGA-UCT modification that detects more state abstractions, improves the MCTS performance by increasing the sample efficiency as more state abstractions lead to more action abstractions, which improve the sample efficiency. This modification only has a minor runtime overhead (see Tab. 1).

**3.** We formulate three new abstraction frameworks, namely, **A**lternating **S**tate **A**nd **S**tate-**A**ction **P**air **A**bstractions (ASASAP), p(runed)-ASAP and **I**deal **P**runing **A**bstractions (IPA) abstractions. Fig. 1 visualizes their hierarchy. In particular, both IPA and ASAP are special cases of p-ASAP and ASASAP. Though p-ASAP already encompasses both IPA and ASAP, we believe that it further helps understand the core principles behind these abstractions and would help categorize future abstractions.

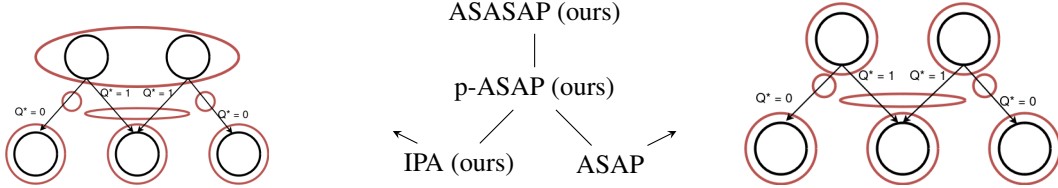

Figure 1: The hierarchy of abstraction frameworks proposed by us that related Ideal Pruning Abstractions (IPA), Abstraction of State-Action Pairs (ASAP), p(runed)-ASAP, and Alternating State And State-Action Pair Abstractions (ASASAP). The leftmost diagram shows an IPA abstraction on an MDP with 5 states, which are black circles that are connected by deterministic actions that are illustrated with black arrows. The red circles show which actions and states IPA groups/abstracts. The same MDP is shown on the right, but with ASAP abstractions that do not manage to detect the equivalence of the two uppermost states.

The paper is structured as follows. Firstly, in **Section** 2, the theoretical groundwork for this paper is laid, in particular, we define our novel ASASAP framework which helps us introduce and classify other abstraction framework such as ASAP or AS from the literature. Next, in **Section** 3, it is first illustrated why OGA-UCT and ASAP-UCT struggle to find state abstractions, after which in Subsection 3 we propose our IPA framework and show how to modify OGA-UCT to approximate IPA abstractions in Subsection 3. We call this modification IPA-UCT. After having described our methodology, the experiment setup is described in **Section** 4. The experimental results and presented and discussed in **Section** 5 where evaluate and analyze IPA-UCT on various domains to verify its capability to boost performance. At the end, in **Section** 6 we briefly summarise our findings and provide an outlook for future work.

## 2 FOUNDATIONS OF AUTOMATIC ABSTRACTIONS

**Problem model and optimization objective:** We use finite MDPs (Sutton & Barto, 2018) as the model for sequential, perfect-information decision-making tasks. We use $\Delta(X)$ to denote the probability simplex of a finite, non-empty set $X$ and $\mathcal{P}(X)$ to denote the power set of $X$.

*Definition:* An *MDP* is a 6-tuple $(S, \mu_0, \mathbb{A}, \mathbb{P}, R, T)$ where the components are as follows:

- $S \neq \emptyset$ is the finite set of states, $T \subseteq S$ is the (possibly empty) set of terminal states, and $\mu_0 \in \Delta(S)$ is the probability distribution for the initial state.

- $\mathbb{A}\colon S \mapsto A$ maps each state $s$ to the available actions $\emptyset \neq \mathbb{A}(s) \subseteq A$ at state $s$ where $|A| < \infty$.

- $\mathbb{P}\colon S \times A \mapsto \Delta(S)$ is the stochastic transition function where we use $\mathbb{P}(s' \,|\, s, a)$ to denote the probability of transitioning from $s \in S$ to $s' \in S$ after taking action $a \in \mathbb{A}(s)$ in $s$.

- $R\colon S \times A \mapsto \mathbb{R}$ is the reward function.

From hereon, let $M = (S, \mu_0, \mathbb{A}, \mathbb{P}, R, T)$ be an MDP. Using the same notation as Anand et al. (2015), we also define $P := \{(s, a) \mid s \in S, a \in \mathbb{A}(s)\}$ as the set of all state-action pairs. The goal is to find an agent $\pi$ that is modelled as a mapping from states to action distributions $\pi\colon S \mapsto \Delta(A)$ such that $\pi$ maximizes the expected episode's return where the (discounted) return for of episode $s_0, a_0, r_0, \ldots, s_n, a_n, r_n, s_{n+1}$ with $s_{n+1} \in T$ is given by $\gamma^0 r_0 + \ldots + \gamma^n r_n$.

**Abstraction frameworks:** Next, we will define a general abstraction framework that includes most of the here-presented abstraction algorithms and capture their core working principle. We bluntly call this framework **A**lternating **S**tate **A**nd **S**tate-**A**ction-**P**air **A**bstractions (ASASAP) whose purpose is to unify parts of the abstraction zoo. The idea of ASASAP is to alternately construct a state abstraction from a state-action-pair abstraction and vice versa. For our purposes, we simply define state and action abstractions as equivalence relations (equivalently partitions) of the state or action space. In the supplementary materials in Section A.6, we show a concrete example of how an ASAP abstraction (a special case of ASASAP) is built.

*Definition*: We call the equivalence relation $\mathcal{H} \subseteq P \times P$ induced by some $n \in \mathbb{N}$, some initial state equivalence relation $\mathcal{E}_0$, mappings $f\colon \mathcal{P}(S \times S) \mapsto \mathcal{P}(P \times P)$ and $g\colon \mathcal{P}(P \times P) \mapsto \mathcal{P}(S \times S)$ to equivalence relations an $ASASAP_{f,g,n,\mathcal{E}_0}$ abstraction if it is of the form

$$\mathcal{H} = \mathcal{H}_n, \tag{1}$$
$$\mathcal{H}_{i+1} = f(\mathcal{E}_i) \qquad\qquad \forall i, \tag{2}$$
$$\mathcal{E}_{i+1} = g(\mathcal{H}_{i+1}) \qquad\qquad \forall i. \tag{3}$$

If additionally $\mathcal{H}$ is invariant to any number of additional applications of $f$ and $g$, then we call it *converged*.

Next, we will present some concrete instances of ASASAP from the literature. Firstly, Jiang et al. (2014) used- and Givan et al. (2003) proposed Automatic State abstractions in UCT (AS-UCT) (the name was given by Anand et al. (2015)), which defines $g_{\text{AS}}(\mathcal{H}_{i+1})$ as grouping states if and only if they have identical legal actions and they are pairwise equivalent:

$$(s_1, s_2) \in g(\mathcal{H}_{i+1}) \iff \mathbb{A}(s_1) = \mathbb{A}(s_2) \land$$
$$\forall a_1 \in \mathbb{A}(s_1)\colon ((s_1, a_1), (s_2, a_1)) \in \mathcal{H}_{i+1}. \tag{4}$$

And any state-action-pair $(s_1, a_1), (s_2, a_2)$ is equivalent i.e. $((s_1, a_1), (s_2, a_2)) \in f(\mathcal{E}_i)$ if and only if the state-action pairs have similar immediate rewards and transition distributions:

$$|R(s_1, a_1) - R(s_2, a_2)| \leq \varepsilon_{\text{a}}$$
$$\text{and } F := \sum_{x \in \mathcal{X}} \left| \sum_{s' \in x} \mathbb{P}(s' \mid s_1, a_1) - \mathbb{P}(s' \mid s_2, a_2) \right| \leq \varepsilon_{\text{t}}, \tag{5}$$

where $\mathcal{X}$ are the equivalence classes of $\mathcal{E}_i$ and $\varepsilon_{\text{t}}, \varepsilon_{\text{a}} \geq 0$. In general, for $\varepsilon_{\text{t}}, \varepsilon_{\text{a}} > 0$, $f(\mathcal{E}_i)$ is not an equivalence relation because transitivity is not guaranteed. Hence, any abstraction algorithms using these, need to be slightly modified. The reason for allowing $\varepsilon_{\text{a}}$ and $\varepsilon_{\text{t}}$ to be greater than 0, is to find more correct abstractions at the cost of potentially abstracting state-action-pairs or states that do not have the same value. The experiments of this paper confirm that this can be beneficial.

To allow for the detection of more symmetries, Anand et al. (2015) proposed ASAP abstractions that are based on Ravindran & Barto (2004) homomorphism condition that does not require there to be a 1-to-1 action match but only a mapping of actions to each other, concretely $g_{\text{ASAP}}(\mathcal{H}_{i+1})$ is defined as

$$(s_1, s_2) \in g_{\text{ASAP}}(\mathcal{H}_{i+1}) \iff$$
$$\forall a_1 \in \mathbb{A}(s_1)\, \exists a_2 \in \mathbb{A}(s_2)\colon ((s_1, a_1), (s_2, a_2)) \in \mathcal{H}_{i+1}$$
$$\forall a_2 \in \mathbb{A}(s_2)\, \exists a_1 \in \mathbb{A}(s_1)\colon ((s_1, a_1), (s_2, a_2)) \in \mathcal{H}_{i+1}. \tag{6}$$

The action abstraction is the same as the previously defined $f$ using $\varepsilon_{\text{t}} = \varepsilon_e = 0$, however, as we will later see, there is nothing that would prevent us from choosing epsilon values greater than zero here.

**Building and using abstractions to enhance search:** Constructing ASAP, AS (or IPA, see Section 3) abstractions until convergence for an entire MDP is oftentimes infeasible, and such a computation would significantly hamper the runtime. Hence, ASAP-UCT (Anand et al., 2015), AS-UCT, OGA-UCT (Anand et al., 2016), and IPA-UCT (see Section 3) build their ASASAP abstraction on the **local-layered MDP** rooted at the state $s_\text{d}$ where the decision has to be made.

*Definition*: The state space of the *layered MDP* of $M$ is $S \times \{0, \ldots, h\}$ where $h \in \mathbb{N}$ is the horizon and if $(s, n)$ is a successor state of $(s', n')$, then $n = n'+1$ and any initial state has $n = 0$. Additional terminal states are $S \times \{h\}$. The *local-layered MDP* rooted at $s_\text{d}$ is the layered MDP of $M$ but with its states, actions, and possible state-action-pair-successors restricted to those present in the current search tree.

In local-layered MDPs, a converged ASAP, AS, or IPA abstraction can be efficiently computed with dynamic programming, where one requires only the abstraction of the previous layer to compute the abstractions for the next. In ASAP-UCT and AS-UCT, an ASAP/AS-like abstraction is built in regular intervals on the current MCTS (for details on MCTS, see Section A.12) search graph using an initial state equivalence relation that groups all terminal states of the same layer, groups all non-fully-expanded nodes of the same layer, and puts all remaining nodes in their own abstract node of size one. The abstraction built by ASAP/AS-UCT differs only to the ASAP/AS abstraction in that non-fully-expanded nodes are never grouped with fully-expanded nodes. This non-grouping condition also applies to OGA-UCT (Anand et al., 2016) and IPA-UCT (see Section 3).

Unlike ASAP-UCT and AS-UCT the successor of ASAP-UCT, OGA-UCT, and its derivatives do not compute their respective abstraction from the ground up but rather attempt to approximate the ASAP (or IPA) abstraction by rebuilding only parts of their current abstraction.

**OGA-UCT for multi-agent settings:** In the experiments, we will also evaluate OGA-based algorithms on board games, which are not MDPs as they feature two players. The only modification needed for OGA is to optimize and keep track of the Q values for the player at the turn at the corresponding node.

**OGA-UCT extensions to high stochasticity settings:** A core weakness of ASAP abstractions is their exactness, which causes them to not be able to deal with stochasticity well. Hence, Anand et al. (2016) directly proposed *pruned OGA* as an improvement to OGA-UCT, which is the same as OGA-UCT except that for the abstraction construction step for each state-action pair with $n$ successors with respective probabilities $p_1, \ldots, p_n$ only those with $p_i > \alpha \cdot \max\{p_1, \ldots, p_n\}$, $\alpha \in [0, 1]$ are considered. Also in this paper, we consider $(\varepsilon_\text{a}, \varepsilon_\text{t})$-OGA (Schmöcker et al., 2025) which is equivalent to OGA-UCT except that one allows for $\varepsilon_\text{a}, \varepsilon_\text{t}$ to be greater than 0. Since this does not induce an equivalence relation, the abstraction construction process has to be slightly modified as detailed by Schmöcker et al. (2025).

**RSTATE-OGA**: Later, when the different OGA variants are experimentally investigated, one ablation that will also be conducted is to test the performance of random state abstractions to ensure that any performance gains due to the usage of abstractions are better than if random abstractions were used. OGA-UCT that uses random state abstractions is called RSTATE-OGA and functions as follows. Whenever a state node $\mathcal{S}$ is visited for the $K$-th time and its current abstract node consists only of itself, then with the probability $p_\text{abs} \in [0, 1]$, $\mathcal{S}$'s abstract node is changed with uniform probability to any of the abstract nodes of the same depth. Initially, at creation, any Q node is its own abstract node.

**Abstraction usage:** Thus far, we have only discussed how to build abstractions but not how to use them. The key mechanism that all here-presented MCTS-based abstraction methods use (e.g. AS-UCT, ASAP-UCT, OGA-UCT, $(\varepsilon_\text{a}, \varepsilon_\text{t})$-OGA, pruned OGA, and IPA-UCT) is only to modify the tree policy by enhancing the UCB value. The UCB value for an action is enhanced by using the aggregated visits and returns of all actions that are part of the same abstract action (i.e., equivalence class). In particular, state abstractions are not used at this stage. These are only needed as an intermediate step to find action abstractions. Only AS-UCT differs slightly from this approach, as it only aggregates the statistics of actions that additionally have the same abstract parent. This is because AS-UCT was originally intended as a state-only abstraction, which is why it did not decouple action and state abstractions.

**Other automatic abstraction algorithms:** A different abstraction paradigm is PARSS by Hostetler et al. (2015) that initially groups all successors of each state-action pair. As the search progresses, this coarse abstraction is refined by repeatedly splitting abstract nodes in half. Another technique is to build, but then fully abandon an abstraction mid-search, a method coined Elastic MCTS by Xu et al. (2023). Though not fully domain-independent, another approach is given by Sokota et al. (2021), who group states based on a domain-specific distance function, and the maximal grouping distance shrinks as the search progresses. While also not in scope of this paper, research effort on abstractions is also dedicated to continuous and/or partially observable problems (Hoerger et al., 2024), and learning-based methods, such as learning and planning on abstract models (Ozair et al., 2021; Kwak et al., 2024; Chitnis et al., 2020), or building abstractions that rely on learned functions (e.g. a value function) (Fu et al., 2023). Research effort has also been dedicated towards automatic abstractions of the transition function, which on an abstract level can be described as pruning certain successors from the transition function (Sokota et al., 2021; Yoon et al., 2008; 2007; Saisubramanian et al., 2017). A more thorough literature overview is given in a survey by Schmöcker & Dockhorn (2025).

## 3 METHOD

In this Section, we will be introducing our novel IPA-UCT algorithm by first showing that ASAP struggles with finding state abstractions. Then we will be introducing a new abstraction framework called IPA, which IPA-UCT tries to approximate. Lastly, we illustrate on a concrete example how the IPA framework detects state equivalences that ASAP does not.

**Why ASAP finds (nearly) no state abstractions:** In the following, we will first give theoretical arguments why ASAP finds few state abstractions, only after which we show experimental evidence to support these claims.

*Theory*: Let us consider a simplified model where two states $s_1, s_2$ with $n$ and $l$ actions respectively are given. Furthermore, assume that each of $s_1$'s and $s_2$'s actions are assigned to an abstract Q node from a pool of $m$ abstract Q nodes with uniform probability. Using elementary combinatorial arguments, the probability $p_{\text{abs}}$ of $s_1$ and $s_2$ being abstracted according to the ASAP framework can be exactly denoted and then upper bounded by

$$p_{\text{abs}} = \frac{\sum_{k=1}^{c:=\min\{n,l,m\}} \binom{m}{k} f(n,k) f(l,k)}{m^{n+l}} \leq \left(\frac{2c}{m}\right)^{n+l} \tag{7}$$

where $f(n,k)$ is the number of surjections from a set of $n$ elements to a set of $k \leq n$ elements (proof is provided in the supplementary materials Section A.7). This shows that once there is a critical amount of possible abstractions $m$, then the probability decays at least exponentially in the number of actions $n$ and $l$. The method IPA that we propose won't depend on $m$.

*Empirical results*: Aside from these theoretical arguments, we empirically measured the abstraction rate for OGA-UCT. The measurements can be seen in the supplementary materials in Tab.2 in the OGA column with $\varepsilon_a = \varepsilon_t = 0$. Clearly, with a few exceptions, nearly no state abstractions are built despite the fact that at least value-equivalent states have to exist in some environments due to symmetry reasons such as Game of Life and SysAdmin. There are two notable exceptions, namely Crossing Traffic and Skills Teaching where standard OGA detects a notable number of state abstractions. However, these are arguably trivial: In Skills Teaching, to simulate the student's learning process, every other turn has only a single action, hence these state abstractions are essentially action abstractions. In Crossing Traffic, once the agent has been hit by an obstacle, the game reaches a non-terminal states in which all the agent's actions have no effect. These trivial dead states are detected.

**The p-ASAP and IPA abstraction frameworks:** First, let us introduce an ASASAP abstraction framework that we call p(runed)-ASAP of which ASAP is a special case. Given a state-action-pair abstraction $\mathcal{H}$, some action pruning function $J \colon S \mapsto \mathcal{P}(A)$ such that $J(s) \subseteq \mathbb{A}(s)$ for all $s$, we can define a symmetric and reflexive (but not necessarily transitive) relation $\sim_J$ with $s_1 \sim_J s_2$ if and

only if

$$\forall a_1 \in J(s_1) \, \exists a_2 \in \mathbb{A}(s_2): \quad ((s_1, a_1), (s_2, a_2)) \in \mathcal{H}, \tag{8}$$

$$\forall a_2 \in J(s_2) \, \exists a_1 \in \mathbb{A}(s_1): \quad ((s_1, a_1), (s_2, a_2)) \in \mathcal{H}. \tag{9}$$

If $\sim_J$ is an equivalence relation, then we call the ASASAP abstraction using $f(\mathcal{H}) = \sim_J$ a p-ASAP abstraction.

ASAP is obtained from p-ASAP by using $J_{\text{ASAP}}(s) = \mathbb{A}(s)$ for all $s$. As illustrated in Section 3, the ASAP framework is practically unable to detect any state abstractions, hence, the goal is to find a $J$ that does as much pruning as possible whilst keeping value-invariance of the resulting abstraction. This is already guaranteed by ASAP because when all actions have an equivalent match, then it is also the case for the optimal action which determines the $V^*$ value.

If one chooses $J^*(s)$ to be the set of optimal actions (i.e., those with the maximal $Q^*$ value), then the maximal number of action pruning is performed whilst ensuring value equivalence of any two abstracted states. Note that $\sim_{J^*}$ is an equivalence relation. We refer to the $p$-ASAP abstraction using $J^*$ as **I**deal-**P**runing-**A**bstractions (IPA) as only those states are grouped that have the same value under optimal play. However, this framework still does not capture all value equivalent states, as two states may be value equivalent but have no ASAP-equivalent actions.

**IPA versus ASAP on Navigation:** In this section, we will demonstrate a motivating example in which the ASAP framework is unable to detect some state abstractions that are encompassed by the IPA framework. Consider the Navigation instance that is illustrated in Fig. 2 and whose definition is given in the supplementary materials A.13. Counterintuitively, the optimal policy is to continuously attempt the straight path $3 \to 8 \to 13 \to 18$ which yields an average return of $-3$. Going around cell 8, either left or right, has a lower average return of $-5$. To decrease the chance of MCTS to take one of these suboptimal paths, it would be of benefit if states 2 and 4 are abstracted as that would imply the actions $3 \to 2$ and $3 \to 4$ could be abstracted too, thus allowing MCTS to average their Q values and therefore decrease this suboptimal-path probability.

However, according to the ASAP framework, states 12 and 14 cannot be abstracted, since the action $14 \to 15$, does not have an ASAP-equivalent action in state 12 (this can be checked using that ASAP-equivalent actions must also be value-equivalent). Consequently, states 7,9 and ultimately 2,4 won't be abstracted. On the contrary, the IPA framework does find all of these abstractions, as going from 12 to 13 or from 14 to 13 are the unique optimal actions which are abstracted since they result in the same ground state.

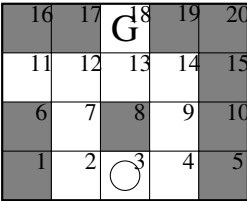

Figure 2: A 5×4 Navigation instance to illustrate an example where the IPA framework (our method) detects the value equivalencies of states 2,4 and 7,9 and 12,14 which cannot be done with ASAP. The circle indicates the initial position, G indicates the goal cell, white cells have a reset probability of 0, and black cells have a reset probability of 0.5.

**IPA-UCT:** Next, we will discuss how the IPA abstraction framework can be integrated with OGA-based methods, which we then call **IPA-UCT**. IPA-UCT will only modify the state abstraction component of OGA; hence, we regard either using the mechanism of pruned OGA or $(\varepsilon_a, \varepsilon_t)$-OGA for the state-action pair abstraction simply as a parametrization of IPA-UCT. We proceed as follows. Firstly, we will introduce an approximation for $J^*$. Since this approximation $J_{\text{UCB}}$ does not induce an equivalence relation, we will show how we can transform it into one such that it can be incorporated into OGA in the supplementary materials in Section A.11. Our idea is to approximate $J^*(s)$ using current search tree information. The approximation $J_{\text{UCB}}$ for $J^*$ for a state $s$ is

$$J_{\text{UCB}}(s) = \{a \in \mathbb{A}(s) \mid \text{UCB}(a) \geq Q_{\max}\} \tag{10}$$

where $Q_{\max} = \max\limits_{a \in \mathbb{A}(s)} Q(s, a)$ is the maximum Q value statistic at node $s$ and $\text{UCB}(a)$ denotes the current UCB value of action $a$ in state $s$. The idea is that, mostly, one cannot tell what the optimal action is; however, given enough visits, one can oftentimes exclude some actions which are almost certainly not optimal. We parametrize this technique by some $\lambda_{\mathrm{p}} \in \mathbb{R} \cup \{\infty\}$ which is used as the exploration constant for the UCB values that are used for this pruning procedure. If one chooses $\lambda_{\mathrm{p}} = 0$, then only those actions are kept that have the current maximum Q value. If one selects $\lambda_{\mathrm{p}} = \infty$, then no pruning takes place, hence $J_{\text{UCB}} = J_{\text{ASAP}}$. Hence, $\lambda_{\mathrm{p}}$ controls the riskiness when building the state abstractions. As will be later shown, the best performances are reached with non-trivial $\lambda_{\mathrm{p}}$ values that have an optimal tradeoff between finding additional correct state abstractions that are built at the cost of faulty new ones. Since the state abstractions now do no longer exclusively depend on the abstractions of the Q nodes, we introduce a recency counter for state nodes as well. The recency count's threshold is set for simplicity to the same value that is used for the Q nodes. As with the Q nodes, whenever that recency counter reaches the threshold, we update the state abstraction.

Though heuristical in nature, IPA-UCT has the same soundness guarantee as OGA-UCT as specified in the following theorem, which is proven in the supplementary materials in Section A.1.

*Soundness theorem:* The abstraction on IPA-UCT's search tree will become sound (i.e. group only states with the same $V^*$ value and state-action pairs with the same $Q^*$ value) almost surely in the iteration limit when using OGA-UCT as the state-action pair abstraction mechanism.

## 4 EXPERIMENT SETUP

In this section, we describe the general experiment setup. Any deviations from this setup will be explicitly mentioned.

**Parameters:** Originally, OGA (Anand et al., 2016) used the absolute value of the abstract Q value as the exploration constant. However, this technique has been improved by the dynamic, scale-independent exploration factor global-std [1]. The global-std exploration constant has the form $C \cdot \sigma$ where $\sigma$ is the standard deviation of the Q values of all nodes in the search tree and $C \in \mathbb{R}^+$ is some fixed parameter. Furthermore, we always use $K = 3$ as the recency counter, which was proposed by Anand et al. (Anand et al., 2016).

**Problem models:** For this paper, we ran our experiments on a variety of MDPs, all of which are either from the International Probabilistic Planning Conference (Grzes et al., 2014), are well-known board games, or are commonly used in the abstraction algorithm literature (Anand et al., 2015; 2016; Hostetler et al., 2015; Yoon et al., 2008; Jiang et al., 2014). All experiments were run on the finite-horizon versions of the considered MDPs with a default horizon of 50 steps and 100 for the board games with a planning horizon of 50 and a discount factor $\gamma = 1$. The board games are zero-sum and transformed into an MDP by inserting standard MCTS with 500 iterations as the opponent. If the reader is not familiar with any of the domains that were used for the experiments, a brief description for each MDP is provided in the supplementary materials in Section A.13.

**Evaluation:** Each data point that we denote in the remaining sections of this paper (e.g. agent returns) is the average of at least 2000 runs. Whenever we denote a confidence interval for a data point, then this is always a confidence interval with a confidence level of 99% provided by $\approx 2.33$ times the standard error. Furthermore, we use a borda-like ranking system to quantify agents' performances; in particular, we use *pairings* and *relative improvement scores*. For details, see supplementary Section A.4.

**Reproducibility:** For reproducibility, we released our implementation (Authors, 2025). Our code was compiled with g++ version 13.1.0 using the -O3 flag (i.e. aggressive optimization).

## 5 EXPERIMENTS

First, we compare the overall performances of IPA-UCT, pruned OGA, and $(\varepsilon_{\mathrm{a}}, \varepsilon_{\mathrm{t}})$-OGA, and RSTATE-OGA (to ensure any performance gains come from non-trivial sources) by computing

---

[1]Citation excluded for double anonymous review process

their pairings and relative improvement for different iteration budgets obtained from the performance values of all $> 20$ considered environments. The parameters we varied are the following: For all methods, we used $C \in \{0.5, 1, 2, 4, 8, 16\}$ For $(\varepsilon_a, \varepsilon_t)$-OGA, we tested $\varepsilon_a \in \{0, \infty\}$, $\varepsilon_t \in \{0, 0.2, 0.4, 0.8\}$, for pruned OGA we used $\alpha \in \{0, 0.1, 0.2, 0.5, 0.75, 1.0\}$, and for RSTATE-OGA we used $p_{\text{abs}} \in \{0.1, 0.2, 0.5, 1.0\}$. We varied the pruning constant $\lambda_p \in \{0, 0.25, 0.5, 1, 2, 4, \infty\}$ where $\lambda_p = \infty$ corresponds to doing no pruning at all i.e. defaulting to standard pruned OGA or $(\varepsilon_a, \varepsilon_t)$-OGA. Each parameter combination was run with 100, 200, 500, and 1000 iterations.

Bar charts 3 compare the pairings and relative improvement scores for the best parameter-combinations for each iteration budget. This shows that using IPA-UCT clearly has better generalization capabilities than not using the modified state abstraction that IPA-UCT introduces, with a sweet spot in performance being the 200 iterations setting where an average 5% performance increase over the best OGA parameter combination can be found. Though the relative improvement scores can go as low as only an average 1% improvement, the pairings score in contrast shows the consistency of this advantage where the average pairings score can go as high as over 20%. IPA-UCT can however, only gain a clear advantage over OGA-based techniques in this generalization setting as the per-environment parameter-optimized yield only minor (if any) improvements except for Cooperative Recon, which we visualized and discuss in the supplementary materials Section A.2. Nonetheless, these results show that IPA-UCT can be a valuable drop-in improvement for OGA-based algorithms, offering a clear advantage when one cannot afford to fine-tune parameters per task. In the next section, we discuss how $\lambda_p$ can be chosen.

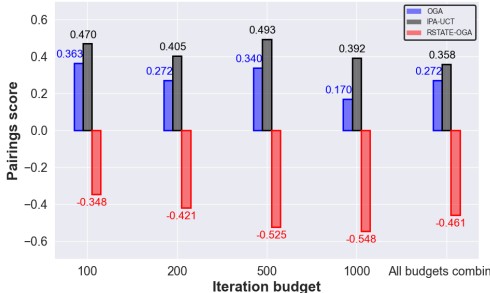 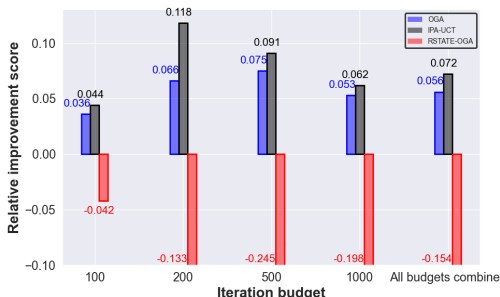

Figure 3: The pairings scores of the best IPA-UCT parameter combination compared to the best parameter combination of RSTATE-OGA, and both pruned OGA and $(\varepsilon_a, \varepsilon_t)$-OGA (summarized as OGA). The overall generalization performance of IPA-UCT for all iteration settings was achieved using $(0, 0.4)$-OGA with $\lambda_p = 1$.

Figure 4: The relative improvement scores of the best IPA-UCT parameter combination compared to the best parameter combination of RSTATE-OGA, and both pruned OGA and $(\varepsilon_a, \varepsilon_t)$-OGA (summarized as OGA). The overall generalization performance of IPA-UCT for all iteration settings was achieved using $(0, 0.2)$-OGA with $\lambda_p = 1$.

**Ablation: Performance as a function of $\lambda_p$:** Next, we investigate the relative performance between the $\lambda_p$ values. Fig. 5 shows the performance curve when varying $\lambda_p$ for the here-considered iteration budgets in terms of the pairings and relative improvement score for all environments (the performance graphs for each individual environment can be found in the supplementary materials in Fig. 8 and Fig. 9. The following observations can be made:

**1)** First and foremost, all curves feature a clear downwards trend as $\lambda_p$ approaches the largest here-considered value 4. All curves have a peak at less than 4. This further validates the positive impact that the UCB-based pruning has on the performance, as the higher the $\lambda_p$ value, the closer IPA-UCT is to standard OGA.

**2)** For both scores, the curves for 500 and 1000 iterations have a single peak, which is either $\lambda_p = 0.5$ or $\lambda_p = 1$, depending on the score type.

**3)** Surprisingly, even for small iteration counts of 100 or 200 iterations, there are still clear peaks which are at either $\lambda_p = 0$ or $\lambda_p = 0.25$ (except for the 200 iterations pairings score). This makes sense as in lower iteration budgets, IPA-UCT needs to be more risk-taking in the abstraction building

as there aren't enough visits to have confidence in the pruning, hence the $\lambda_p$ peaks are at lower values than for the higher iteration budgets.

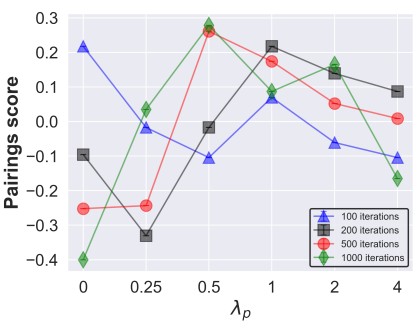

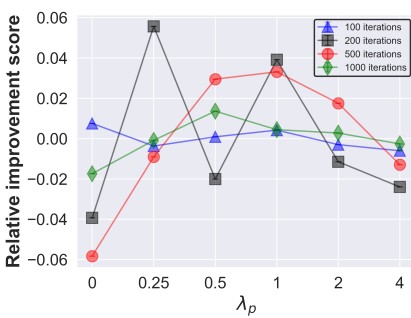

Pairings scores

Relative improvement scores

Figure 5: The pairings and relative improvement scores of different $\lambda_p$ values (when only paired against each other for a fixed iteration budget, i.e. all curves are independent of each other) for different iteration budgets. Though the data points in between $\lambda_p$ weren't measured, we still drew connecting lines for the reader to better differentiate between the course of the scores for the different budgets.

**Alternative pruning methods:** Though we found most success with the proposed $J_{\text{UCB}}$ function of pruning actions that relies on the UCB and Q values only, we also conducted preliminary experiments on two different approaches that are described in the following, and whose performances are shown in Tab. 4. However, both approaches performed worse than $J_{\text{UCB}}$, whose downsides we will briefly cover, and why we ultimately presented IPA-UCT instead of these.

*Confidence-based pruning*: For this method, we kept track of a confidence interval with confidence level $p_c \in [0, 1]$ for each Q value. We then used $J_{\text{conf}}$ which prunes all actions at a state $s$ whose upper confidence bound are lower than the highest lower bound. We call this method *CONF-UCT*. In our observation, the Q values were much too noisy to perform any meaningful pruning.

*Hard pruning*: For this method $J_{\text{top}}$, one only keeps the best $n_{\text{matches}}$ actions when ordered by their current Q value. To avoid the risk of faulty prunings, $J_{\text{top}} = J_{\text{ASAP}}$ if the node has less than $n_{\text{min}}$ visits. This method, which we name *TOPN-UCT*, performed nearly on PAR with $J_{\text{UCB}}$, however, it has two parameters to configure rather than just one.

## 6 CONCLUSION AND FUTURE WORK

In this paper, we first generalized the abstraction framework ASAP by Anand et al. (2015) by introducing p-ASAP and ASASAP. Next, we showed both empirically and theoretically that OGA-UCT effectively finds no state abstractions. We proposed IPA-UCT to alleviate this issue and showed that IPA-UCT yields a consistent performance improvement over OGA-based methods.

One limitation of IPA is that there is no single $\lambda_p$ value that performs well for all environment. One avenue for future work is to automatically detect the correct value for $\lambda_p$. While for some environments $\lambda_p = 0$ is best, this can be harmful to others. Furthermore, some environments prefer neither $\lambda_p = \infty$ (i.e. no pruning) nor $\lambda_p = 0$ but rather some value in between. Also, IPA-UCT is clearly not optimal in that still many state abstractions, especially those arising due to symmetry (e.g., in Game of Life or SysAdmin), are not detected because as OGA-UCT, IPA-UCT relies on the detection of action abstractions. We believe that this near-exact abstraction that is being built in IPA-UCT and OGA-UCT is not the path forward to resolve this issue, as it requires too many Q nodes to have sampled nearly all their possible outcomes which is mostly infeasible when the stochasticity has more than binary outcomes. A new automatic abstraction paradigm is required. Lastly, since IPA-UCT is a modification of OGA-UCT is suffers from the same limitation in that a directed acyclic search graph is required for any abstractions to be detected because if no two state-action-pairs result in the same state, then no action abstraction can be built.

## 7 REPRODUCIBILITY STATEMENT

In our experiment setup, we have a subsection called *Reproducibility* in which we provide a download link to the full codebase used for this project as well as compilation details. The codebase contains an elaborate README detailing the steps to reproduce the experiments.

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

# A  SUPPLEMENTARY MATERIALS

## A.1  PROOF OF SOUNDNESS THEOREM

In IPA-UCT using standard OGA-UCT, every state-action pair of the MDP will be expanded, and its visits converge in probability to $\infty$ due to UCB's exploration term. Hence, in the limit, every state-action pair successor will have also been sampled almost surely. Next, assuming that this is the case, one shows inductively, starting from the bottom layer, that the built abstraction will become sound almost surely.

**Induction start**: The fully expanded search tree's bottom layer contains only terminal states, which are grouped by default. This abstraction is sound as terminal states have a $Q^*$ value of 0.

**Induction step**: Assume that the state abstraction at layer $L + 1$ becomes sound almost surely. Consequently, the state-action pairs' Q-values in layer $L$ will converge in probability to their $Q^*$ values as UCB is used as the tree-policy. Also, independent of the state-action pair's Q-values, their abstractions almost surely become sound as they are built with the standard ASAP rules. The $V^*$ value of any state is defined as the maximum $Q^*$ value of its actions. Since the Q-values of the state-action pairs in layer $L$ converge in probability to their $Q^*$ values, the Q value of any optimal action $a^*$ for any state $s$ at layer $L$ and therefore its UCB value will almost surely be greater than the Q value of any suboptimal action of $s$. Therefore, all the optimal actions of $s$ will almost surely never be pruned simultaneously. Hence, the state abstraction at layer $L$ will also almost surely become sound. $\qquad\square$

## A.2 PARAMETER-OPTIMIZED PERFORMANCES

Fig. 6 and Fig. 7 compare the parameter-optimized performances of IPA-UCT, pruned OGA and $(\varepsilon_\mathrm{a}, \varepsilon_\mathrm{t})$-OGA (summarized simply as OGA) as well as RSTATE-OGA using the same parameters as in the main experimental section 5.

The key observation that can be made is that IPA-UCT has only limited use in improving the parameter-optimized, as the gains (if any) are only marginal and could be explained due to noise, except for the Cooperative Recon environment where IPA-UCT can a clear advantage. In Manufacturer, Racetrack, Sailing Wind, Tamarisk, Connect 4, Pylos, and Othello, there seems to be a significant, however extremely small gain.

Though IPA-UCT will show more promise in the generalization experiments presented in the main section, the reasons for the negligible impact of IPA in this setting will be discussed, which can be attributed to three criteria that have to be satisfied for IPA-UCT to have a significant impact, which altogether can be quite rare.

1. The domain must have a small action space. For $J_\mathrm{UCB}$ to prune an action, it must have a low enough exploration term, which shrinks only with the number of visits. If there are too many actions, the visits will be spread too much. This explains the lack of impact in Academic Advising, Game of Life, or SysAdmin. Of course, using $\lambda_\mathrm{p} = 0$ does pruning even with no visits; however, we consider performance gains by pruning under such uncertainty as simply lucky.

2. Ultimately, IPA (as well as ASAP) requires state-action pair abstractions to bootstrap off of. Hence, if almost no action abstractions are found, then IPA cannot detect any state abstractions. Due to its extremely high stochasticity, this is the case for Earth Observation where with reasonable $\varepsilon_\mathrm{t}$ values, no action abstractions are found.

3. There must be state equivalences in the first place. Some environments like Sailing Wind or the here-considered Navigation instance feature almost no state-equivalences (for reference, check their corresponding abstraction rate Tab. 3)

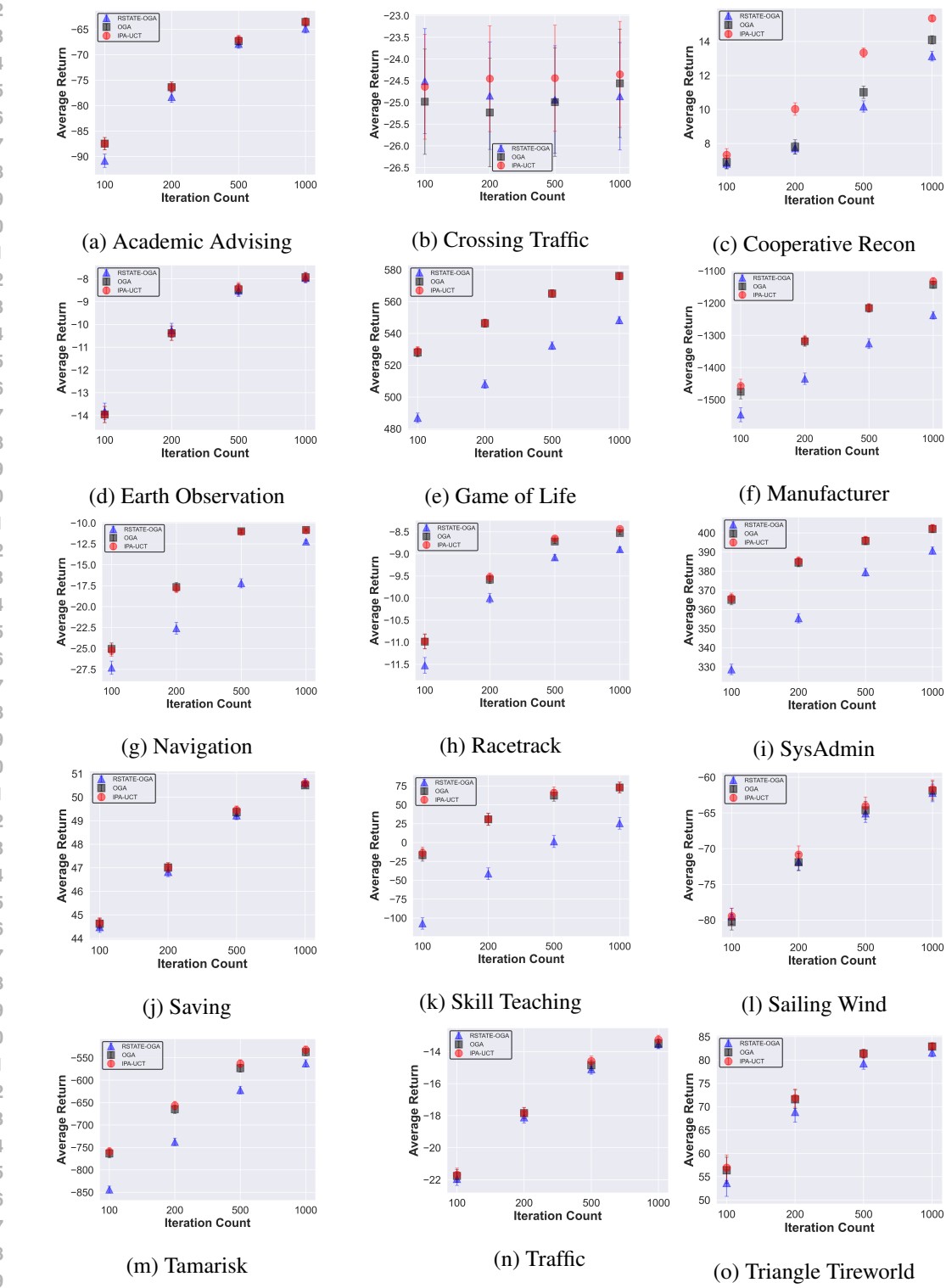

Figure 6: The performance graphs for single-agent problems of in dependence of the MCTS iteration count of the parameter optimized versions of IPA-UCT versus RSTATE-OGA versus pruned OGA and $(\varepsilon_a, \varepsilon_t)$-OGA (summarized as OGA).

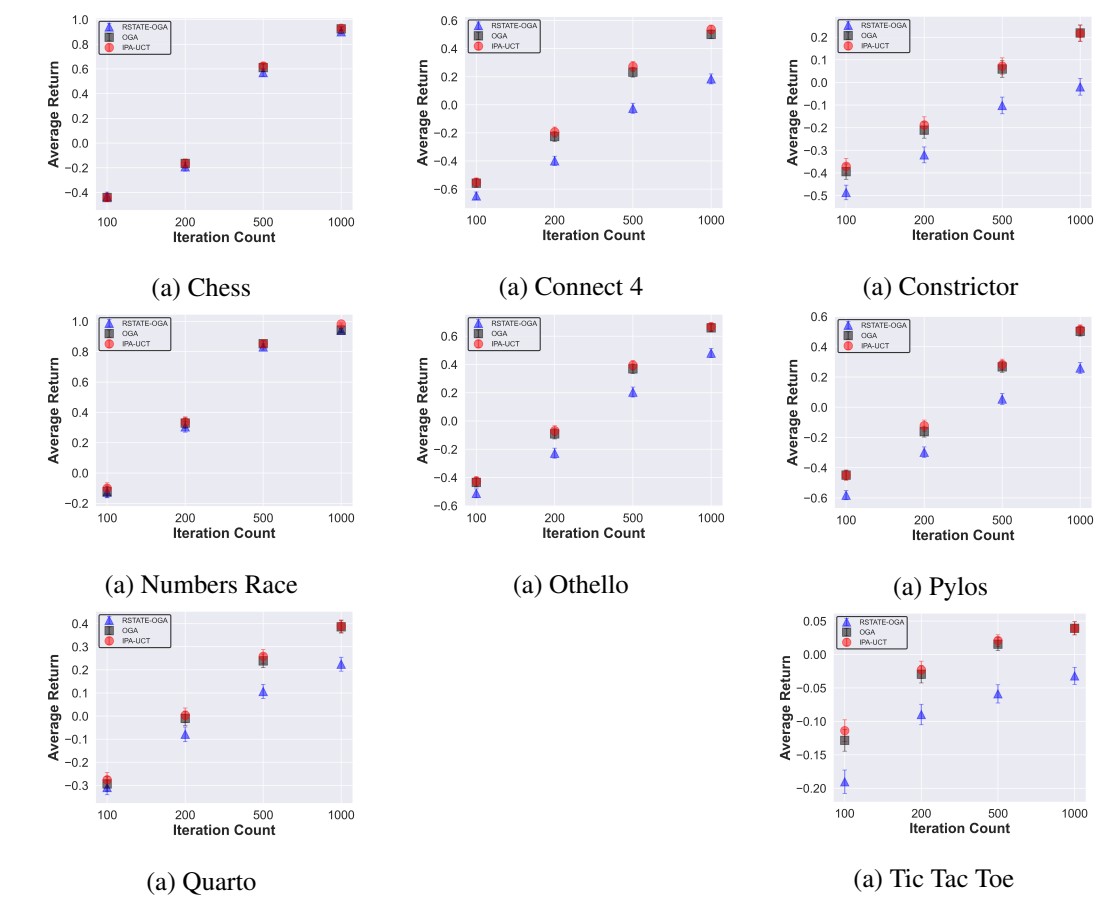

(a) Chess

(a) Connect 4

(a) Constrictor

(a) Numbers Race

(a) Othello

(a) Pylos

(a) Quarto

(a) Tic Tac Toe

Figure 7: The performance graphs for multi-agent problems of in dependence of the MCTS iteration count of the parameter optimized versions of IPA-UCT versus RSTATE-OGA versus pruned OGA and $(\varepsilon_a, \varepsilon_t)$-OGA (summarized as OGA).

## A.3 PARAMETER-OPTIMIZED PERFORMANCE SPLIT BY $\lambda_P$ VALUES

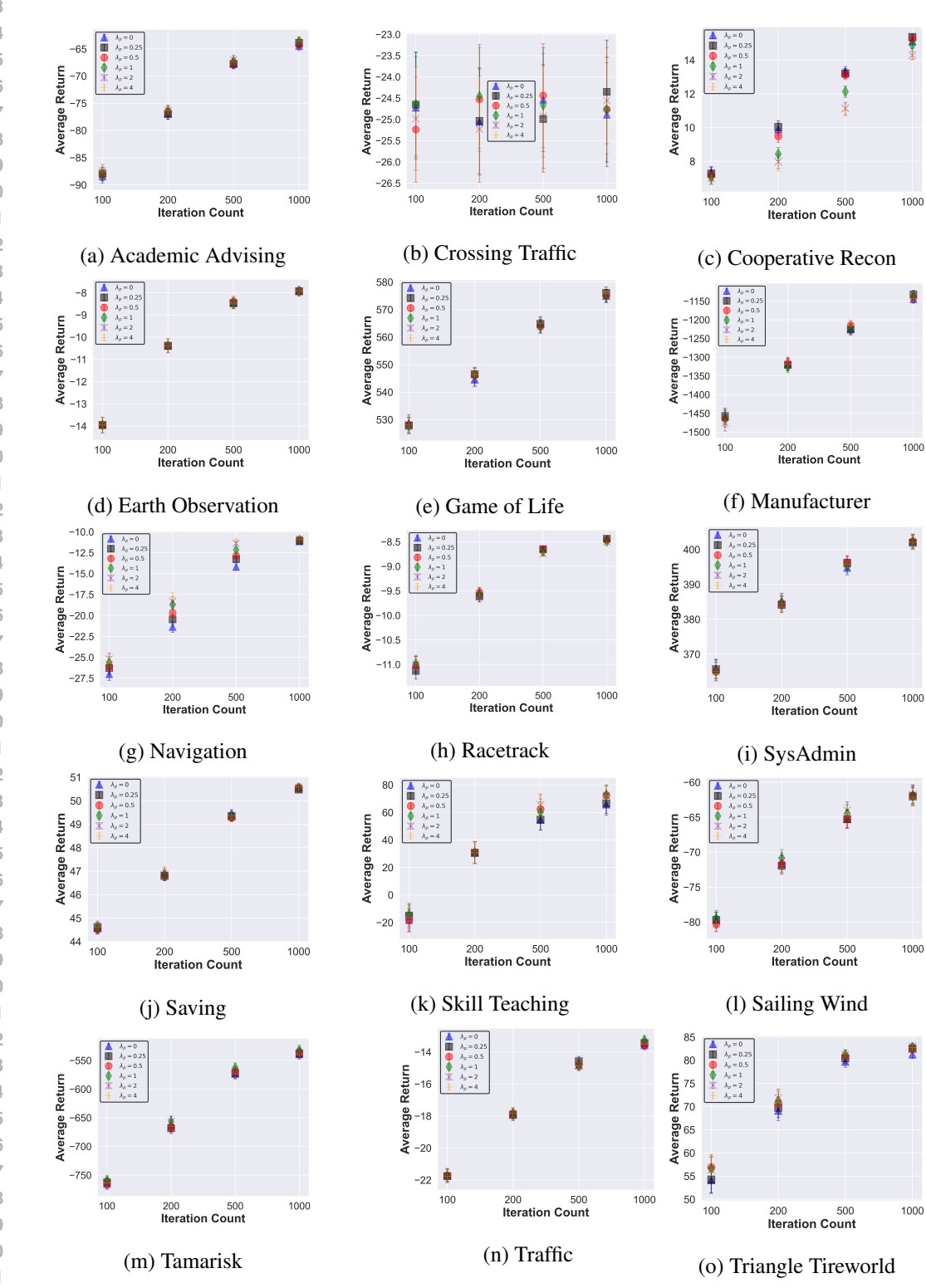

Figure 8: The performance graphs for single-agent problems in dependence of the iteration budget and $\lambda_p$.

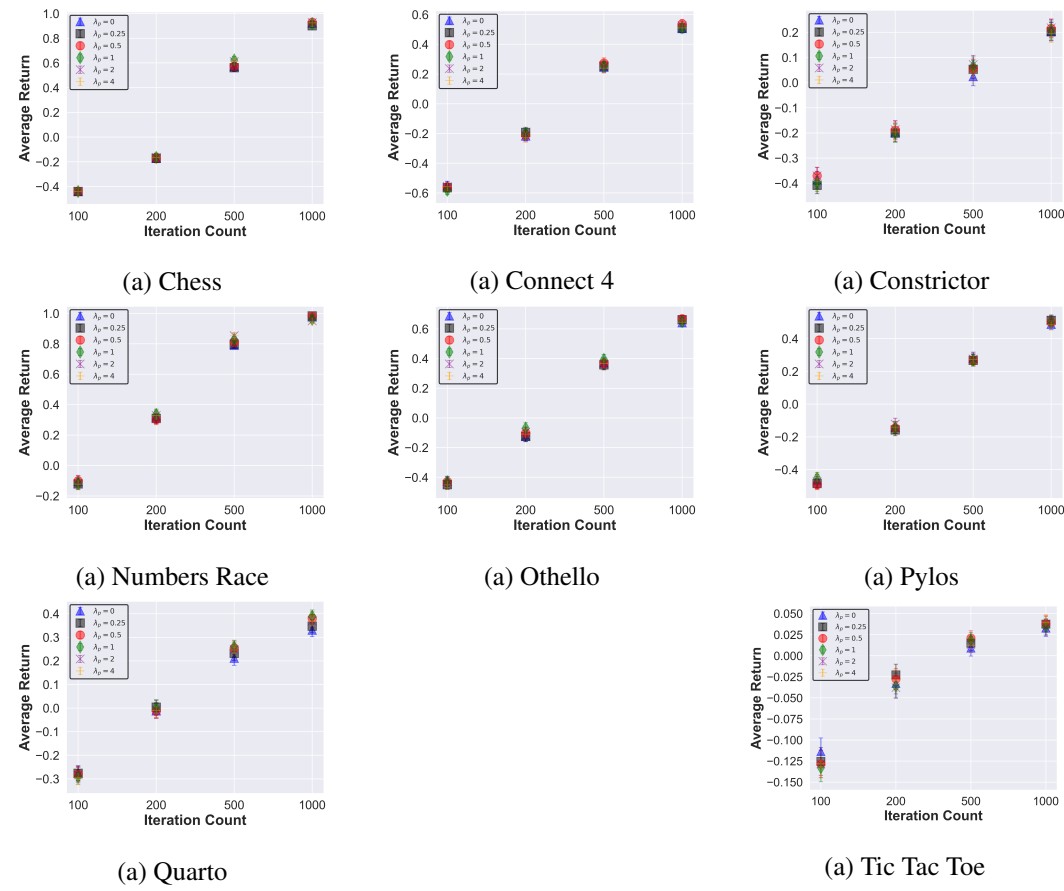

Figure 9: The performance graphs for multi-agent problems in dependence on the iteration budget and $\lambda_p$.

## A.4 DEFINITION OF THE RELATIVE IMPROVEMENT AND PAIRINGS SCORE

In the main experimental section, we evaluated IPA-UCT with respect to the relative improvement and pairings score, which are formalized here. While the pairings score is calculated by summing over the number of tasks where some agent performed better than another, the relative improvement score also takes the percentage of the improvement into account; however, it is prone to outliers. Hence, we considered both scores to paint the full picture.

Concretely, let $\{\pi_1, \ldots, \pi_n\}$ be $n$ agents (e.g., concrete parameter settings) where each agent was evaluated on $m$ tasks (e.g. a given MCTS iteration budget and an environment) where $p_{i,k} \in \mathbb{R}$ denotes the performance of agent $\pi_i$ on the $k$-th task.

**Definition:** The *pairings score matrix* $M \in \mathbb{R}^{n \times n}$ is defined as

$$M_{i,j} = \frac{1}{m-1} \sum_{1 \le k \le m} \text{sgn}(p_{i,k} - p_{j,k}) \tag{11}$$

where sgn is the signum function. The *pairings score* $s_i \le i \le n$ is given by

$$s_i = \frac{1}{n-1} \sum_{1 \le l \le n, l \ne i} M_{i,l}. \tag{12}$$

**Definition** The *relative improvement matrix* $M \in \mathbb{R}^{n \times n}$ is defined as

$$M_{i,j} = \frac{1}{m-1} \sum_{1 \le k \le m} \frac{p_{i,k} - p_{j,k}}{\max(|p_{i,j}|, |p_{j,k}|)} \tag{13}$$

and the *relative improvement score* $s_i \leq i \leq n$ is given by

$$s_i = \frac{1}{n-1} \sum_{1 \leq l \leq n, l \neq i} M_{i,l}. \tag{14}$$

## A.5 RUNTIME MEASUREMENTS

Tab. 1 lists the average decision-making times for each environment of IPA-UCT compared to OGA-UCT for 100 and 2000 iterations on states sampled from a distribution induced by random walks. This shows that while UCT adds only a minor overhead, despite having to execute more UCB evaluations. In particular, we are using highly optimized environment implementations that could be the runtime bottleneck in more complex environments.

Table 1: Average decision-making times of IPA-UCT versus OGA-UCT in milliseconds for 100 and 2000 iterations. This data was obtained using an Intel(R) Core(TM) i5-9600K CPU @ 3.70GHz. The data shows a median runtime overhead of $\approx 5\%$ for 100 iterations and $\approx 9\%$ for 2000 iterations.

| Domain | IPA-UCT-100 | OGA-UCT-100 | IPA-UCT-2000 | OGA-UCT-2000 |
|---|---|---|---|---|
| Academic Advising | 2.22 | 2.01 | 164.63 | 125.61 |
| Cooperative Recon | 4.14 | 3.91 | 267.31 | 232.49 |
| Crossing Traffic | 2.85 | 2.62 | 382.01 | 378.96 |
| Connect4 | 1.77 | 1.69 | 112.21 | 98.94 |
| Chess | 18.01 | 18.40 | 454.55 | 421.35 |
| Constrictor | 4.96 | 4.71 | 347.41 | 316.53 |
| Earth Observation | 7.61 | 7.92 | 367.06 | 345.02 |
| Game of Life | 4.14 | 4.02 | 273.46 | 260.22 |
| Manufacturer | 10.46 | 10.75 | 332.33 | 323.48 |
| Navigation | 2.57 | 2.34 | 104.53 | 82.99 |
| NumbersRace | 2.26 | 1.33 | 1012.79 | 876.50 |
| Othello | 8.18 | 7.77 | 328.30 | 333.46 |
| Pylos | 4.78 | 4.84 | 229.06 | 206.96 |
| Quarto | 2.96 | 2.89 | 226.28 | 219.62 |
| Racetrack | 1.60 | 1.46 | 85.47 | 82.78 |
| Sailing Wind | 2.23 | 2.15 | 185.44 | 169.06 |
| Saving | 1.48 | 1.37 | 249.19 | 246.40 |
| Skills Teaching | 3.95 | 4.08 | 262.31 | 218.11 |
| SysAdmin | 1.90 | 1.81 | 173.24 | 156.65 |
| Tamarisk | 2.94 | 2.87 | 145.57 | 134.80 |
| Traffic | 3.94 | 3.81 | 171.77 | 167.41 |
| Triangle Tireworld | 4.43 | 3.85 | 143.73 | 125.15 |
| Tic Tac Toe | 1.06 | 0.98 | 54.46 | 47.35 |

## A.6 ASAP ABSTRACTION EXAMPLE

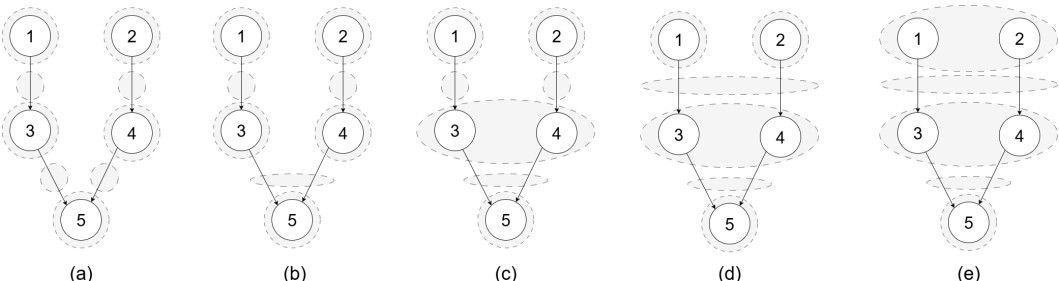

Figure 10: A showcase of how the ASAP abstraction framework, which itself is a special case of ASASAP abstractions, would detect equivalences in the following 5-state MDP. Each node represents a state, and arrows represent deterministic actions with the same immediate reward of 0. The dotted ovals represent abstractions. Initially, in (a), all states and state-action pairs are in their own singleton abstract node. Then, in (b), the next state-action pair abstraction is constructed (the application of function $f$ from Section 2) from this initial state abstraction, which groups the actions of nodes 3 and 4 because they have the same immediate reward and the same transition distribution. From this state-action pair abstraction, the next state abstraction is constructed in (c), (the application of function $g$ from Section 2), which groups nodes 3 and 4 because they have the same set of abstract state-action pairs. Then again, in (d), the next state-action pair abstraction is constructed, which also groups the actions from nodes 1 and 2 because they have the same abstract successor. Then a state abstraction is constructed again in (e), which groups states 1 and 2. Then further applications of $f$ or $g$ would have no effect, hence this abstraction is converged.

## A.7 PROOF OF STATE-ABSTRACTION THEOREM

Here, we will prove the following theorem from the main section:

**Theorem:** Assume $s_1, s_2$ are two states with $n$ and $l$ actions respectively. Furthermore, assume that each of $s_1$'s and $s_2$'s actions is assigned to an abstract Q node from a pool of $m$ abstract Q nodes with uniform probability. The probability $p_{\text{abs}}$ of $s_1$ and $s_2$ being abstracted according to the ASAP framework can be exactly denoted and then upper bounded by

$$p_{\text{abs}} = \frac{\sum_{k=1}^{c:=\min\{n,l,m\}} \binom{m}{k} f(n, k) f(l, k)}{m^{n+l}} \leq \left(\frac{2c}{m}\right)^{n+l} \tag{15}$$

where $f(n, k)$ is the number of surjections from a set of $n$ elements to a set of $k \leq n$ elements.

**Proof:** Let us denote $s_1$'s actions by $A = \{a_1, \ldots, a_n\}$ and $s_2$'s actions by $B = \{b_1, \ldots, b_l\}$. The set of abstract nodes is denoted as $\mathbb{A}_1, \ldots, \mathbb{A}_m$. We denote the abstraction that has uniformly been assigned to an action $c \in \{a_1, \ldots, a_n, b_1, \ldots, b_l\}$ by abs($c$). Using the ASAP framework definition, it holds that

$$p_{\text{abs}} = \mathbb{P}[\{\text{abs}(a_i) \mid 1 \leq i \leq n\} = \{\text{abs}(b_i) \mid 1 \leq i \leq l\}]. \tag{16}$$

Since by assumption the abstraction assignment is uniform, $p_{\text{abs}}$ can be denoted as the ratio of abstraction assignments for $s_1$ and $s_2$ that result in the same set of abstract nodes divided by all possible abstraction assignments. Furthermore, assignments that result in the same set of abstract nodes can be split by the size of that abstract node set. Hence,

$$p_{\text{abs}} = \frac{\sum_{k=1}^{c:=\min(n,l,m)} |\{f : A \mapsto X, \; g : B \mapsto X \mid f, g \text{ surjective}, \; X \subseteq \mathbb{A}, \; |X| = k\}|}{m^{n+l}}$$

$$= \frac{\sum_{k=1}^{c} \binom{m}{k} f(n, k) f(l, k)}{m^{n+l}} \tag{17}$$

where $\mathbb{A} = \{\mathbb{A}_1, \ldots, \mathbb{A}_m\}$. This proves the first part of this theorem. Next, using that $f(n, k) \leq k^n$ yields

$$\sum_{k=1}^{c} \binom{m}{k} f(n, k) f(l, k) \leq \sum_{k=1}^{c} \binom{m}{k} k^{n+l} \leq c^{n+l} \sum_{k=1}^{c} \binom{m}{k} \leq c^{n+l} 2^c \leq (2c)^{n+l} \quad (18)$$

from which the theorem directly follows. □

## A.8 NUMBER OF STATE ABSTRACTIONS BUILT BY OGA AND IPA

Table 2: Comparison of abstraction statistics for different state abstractions and models to show that OGA almost never finds any state abstractions in contrast to our method IPA. Each column denotes the measured ratio of size one state abstractions to the number of total abstractions (excluding trivial abstractions, i.e. those that group all terminal states or size-one abstractions that did not yet receive an update). Hence, the value $1.00$ corresponds to no non-trivial state abstractions, while a value close to $0$ means that almost all states are grouped into node abstract node. The states whose statistics were averaged come from the state distribution of standard OGA-UCT (see Section 4). For IPA-UCT, we used $\lambda_p = 0$. The results were averaged from 100 episodes each. The epsilon values $\varepsilon_t$ denote the transition function threshold defined in Section 2.

| Domain | $\varepsilon_t = 0$ | | $\varepsilon_t = 0.4$ | |
|---|---|---|---|---|
| | OGA | IPA | OGA | IPA |
| Academic Advising | 1.00 | 1.00 | 1.00 | 0.96 |
| Crossing Traffic | 0.50 | 0.55 | 0.50 | 0.55 |
| Cooperative Recon | 1.00 | 0.88 | 0.99 | 0.82 |
| Connect4 | 0.99 | 0.95 | 0.99 | 0.95 |
| Constrictor | 0.99 | 0.99 | 0.99 | 0.99 |
| Earth Observation | 1.00 | 1.00 | 0.94 | 0.92 |
| Game of Life | 1.00 | 1.00 | 0.96 | 0.91 |
| Manufacturer | 1.00 | 1.00 | 1.00 | 0.93 |
| Navigation | 1.00 | 0.95 | 1.00 | 0.90 |
| NumbersRace | 1.00 | 0.91 | 1.00 | 0.91 |

| Domain | $\varepsilon_t = 0$ | | $\varepsilon_t = 0.4$ | |
|---|---|---|---|---|
| | OGA | IPA | OGA | IPA |
| Othello | 0.98 | 0.98 | 0.98 | 0.98 |
| Pylos | 0.97 | 0.96 | 0.97 | 0.96 |
| Quarto | 0.98 | 0.95 | 0.98 | 0.95 |
| Racetrack | 1.00 | 0.93 | 1.00 | 0.93 |
| Sailing Wind | 1.00 | 0.99 | 1.00 | 0.97 |
| Skills Teaching | 0.79 | 0.80 | 0.70 | 0.73 |
| SysAdmin | 1.00 | 0.99 | 1.00 | 0.94 |
| Tamarisk | 1.00 | 1.00 | 1.00 | 0.97 |
| Traffic | 1.00 | 1.00 | 1.00 | 1.00 |
| Triangle Tireworld | 0.99 | 0.93 | 0.99 | 0.92 |

## A.9 RATIO OF VALUE EQUIVALENCES

Table 3: A list of the ratios of local state pairs and action pairs that are value-equivalent to explain why IPA did not improve the performance in Triangle Tireworld or Sailing Wind because these environments contain very few value-equivalent states. These ratios were determined by randomly sampling $10^5$ states. We then applied $i \in \{1, 2, 3\}$ random actions to each state and a copy of each state. We then counted how many times out of these $10^5$ states, the resulting states after applying $i$ actions, had the same value or the same Q-value for the $i$-th action. We denote these ratios by $V_{abs}(i)$ and $Q_{abs}(i)$. Hence, a ratio of $1.00$ would mean that all states in a search tree layer have the same optimal value, while a ratio of $0.00$ means that no two states have the same $V^*$ value.

| Model | $V_{abs}(0)$ | $Q_{abs}(0)$ | $V_{abs}(1)$ | $Q_{abs}(1)$ | $V_{abs}(2)$ | $Q_{abs}(2)$ |
|---|---|---|---|---|---|---|
| Crossing Traffic | 0.83 | 0.89 | 0.84 | 0.88 | 0.85 | 0.88 |
| Navigation | 0.05 | 0.05 | 0.05 | 0.13 | 0.05 | 0.12 |
| Racetrack | 0.38 | 0.53 | 0.37 | 0.42 | 0.34 | 0.37 |
| Sailing Wind | 0.04 | 0.01 | 0.03 | 0.01 | 0.03 | 0.01 |
| Skill Teaching | 0.29 | 0.11 | 0.17 | 0.11 | 0.06 | 0.04 |
| Triangle Tireworld | 0.03 | 0.67 | 0.03 | 0.61 | 0.03 | 0.59 |

## A.10 ALTERNATIVE ACTION PRUNING METHODS STATS

The parameter-optimized performances for the discarded alternative methods CONF-UCT and TOPN-UCT varying $p_c \in \{0.1, 0.25, 0.5, 0.75, 0.9, 0.95\}$ and $n_{min} \in \{0, 20, 100\}$, $n_{matches} \in \{1, 2, 3\}$, and $\varepsilon_t \in \{0, 0.4\}$, $\varepsilon_a = 0$.

Table 4: Average returns and 99% confidence interval for OGA-UCT, TOPN-UCT, and CONF-UCT to show that alternatives to the UCB-based pruning mechanism in IPA-UCT perform slightly worse than IPA-UCT.

| | CONF-UCT | OGA-UCT | RSTATE-OGA | TOPN-UCT |
|---|---|---|---|---|
| Academic Advising | $-65.2 \pm 0.3$ | $-65.3 \pm 0.3$ | $-65.6 \pm 0.3$ | $\mathbf{-65.2 \pm 0.3}$ |
| CaptureTheFlag | $\mathbf{0.00 \pm 0.00}$ | $\mathbf{0.00 \pm 0.00}$ | $\mathbf{0.00 \pm 0.00}$ | $\mathbf{0.00 \pm 0.00}$ |
| Connect4 | $0.22 \pm 0.01$ | $0.22 \pm 0.01$ | $-0.06 \pm 0.02$ | $\mathbf{0.23 \pm 0.02}$ |
| Constrictor | $0.15 \pm 0.02$ | $0.15 \pm 0.02$ | $-0.04 \pm 0.02$ | $\mathbf{0.16 \pm 0.02}$ |
| Cooperative Recon | $12.0 \pm 0.1$ | $12.0 \pm 0.1$ | $11.6 \pm 0.1$ | $\mathbf{12.6 \pm 0.1}$ |
| Crossing Traffic | $-26.1 \pm 0.4$ | $-26.1 \pm 0.4$ | $-26.0 \pm 0.4$ | $\mathbf{-25.6 \pm 0.4}$ |
| Earth Observation | $-8.20 \pm 0.08$ | $-8.19 \pm 0.08$ | $\mathbf{-8.10 \pm 0.08}$ | $-8.19 \pm 0.08$ |
| Game of Life | $572.7 \pm 0.7$ | $572.0 \pm 0.7$ | $\mathbf{572.8 \pm 0.7}$ | $572.8 \pm 0.7$ |
| KillTheKing | $\mathbf{0.02 \pm 0.00}$ | $0.02 \pm 0.01$ | $-0.00 \pm 0.01$ | $0.02 \pm 0.01$ |
| Manufacturer | $-1239.8 \pm 4.0$ | $-1244.5 \pm 4.1$ | $-1241.1 \pm 4.0$ | $\mathbf{-1238.0 \pm 4.0}$ |
| Navigation | $-3.25 \pm 0.02$ | $-3.27 \pm 0.02$ | $-3.50 \pm 0.03$ | $\mathbf{-3.14 \pm 0.03}$ |
| NumbersRace | $0.79 \pm 0.01$ | $0.79 \pm 0.01$ | $0.70 \pm 0.01$ | $\mathbf{0.79 \pm 0.01}$ |
| Othello | $0.23 \pm 0.01$ | $0.23 \pm 0.02$ | $0.07 \pm 0.02$ | $\mathbf{0.24 \pm 0.02}$ |
| Pylos | $0.23 \pm 0.01$ | $0.22 \pm 0.02$ | $-0.03 \pm 0.02$ | $\mathbf{0.23 \pm 0.02}$ |
| Quarto | $-0.27 \pm 0.01$ | $-0.26 \pm 0.02$ | $-0.37 \pm 0.02$ | $\mathbf{0.05 \pm 0.02}$ |
| Racetrack | $-8.76 \pm 0.01$ | $-8.78 \pm 0.01$ | $-9.05 \pm 0.02$ | $\mathbf{-8.50 \pm 0.01}$ |
| Sailing Wind | $-33.4 \pm 0.2$ | $-33.6 \pm 0.2$ | $\mathbf{-33.3 \pm 0.2}$ | $-33.5 \pm 0.2$ |
| Skills Teaching | $\mathbf{62.9 \pm 2.5}$ | $\mathbf{62.9 \pm 2.5}$ | $27.0 \pm 2.6$ | $\mathbf{62.9 \pm 2.5}$ |
| SysAdmin | $400.4 \pm 0.6$ | $399.9 \pm 0.6$ | $400.9 \pm 0.6$ | $\mathbf{401.1 \pm 0.6}$ |
| Tamarisk | $\mathbf{-549.7 \pm 2.6}$ | $-551.4 \pm 2.7$ | $-550.9 \pm 2.7$ | $-550.6 \pm 2.7$ |
| Traffic | $-14.2 \pm 0.1$ | $-14.2 \pm 0.1$ | $-14.2 \pm 0.1$ | $\mathbf{-14.1 \pm 0.1}$ |
| Triangle Tireworld | $\mathbf{82.8 \pm 0.2}$ | $82.5 \pm 0.3$ | $82.0 \pm 0.3$ | $82.6 \pm 0.2$ |

## A.11 How IPA-UCT transforms $J_{\mathrm{UCB}}$ into an equivalence relation

Since $J_{\mathrm{UCB}}$ does not induce an equivalence relation, we cannot simply place any two states $s_1, s_2$ such that $s_1 \sim_{J_{\mathrm{UCB}}} s_2$ into the same abstract node. We will handle this similarly to how we handled the epsilon greater than zero case for the state-action-pairs (i.e. $(\varepsilon_{\mathrm{a}}, \varepsilon_{\mathrm{t}})$-OGA). Again, the aim of the following heuristic is to produce an equivalence relation for states whilst creating as big and stable as possible abstract nodes.

Each abstract state node now also keeps track of its representative, which is one of its original nodes. Furthermore, it is assigned a unique and constant ID at its creation. At its creation, an abstract node is assigned an ID equal to the total number of abstract state nodes that have been created so far. Whenever a state node $s$ is updated, and it is either its own abstract node's representative or when using $J_{\mathrm{UCB}}(s)$ and the value of $J_{\mathrm{UCB}}$ of $s$ at the last time it was updated, the equations 8, 9 do not hold, then the abstract node of $s$ is updated by choosing the largest abstract node (tie breaks by using the ID) with a representative $s'$ such that $s \sim_{J_{\mathrm{UCB}}} s'$. In case this leads to a different abstract node than the current one of $s$, a new representative for the old abstract node is chosen at random.

## A.12 Monte Carlo Tree Search

All here-presented abstraction algorithms rely on Monte Carlo Tree Search (MCTS) which we are going to describe now in detail. Let $M$ be a finite-horizon MDP. On a high level, MCTS repeatedly samples trajectories starting at some state $s_0 \in S$ where a decision has to be made until a stopping criterion is met. The final decision is then chosen as the action at $s_0$ with the highest average return. In contrast to a pure Monte Carlo search, MCTS improves subsequent trajectories by building a tree (or, in our case, a directed acyclic graph) from a subset of the states encountered in the last iterations, which is then exploited. In contrast to pure Monte Carlo search, MCTS is guaranteed to converge to the optimal action.

An MCTS directed acyclic graph is made of two components. Firstly, the state nodes, that represent states and Q nodes that represent state action pairs. Each state node, saves only its children which

are a set of Q nodes. Q nodes save both its children which are state nodes and the number of and the sum of the returns of all trajectories that were sampled starting at the Q node.

Initially, the MCTS search graph consists only of a single state node representing $s_0$. Until the iteration budget is exhausted, the following steps are repeated.

1. **Selection phase**: Starting at the root node, MCTS first selects a Q node according to the so-called *tree policy*, which may use the nodes' statistics, and then samples one of the Q node's successor states. If either a terminal state node, a state node with at least one non-visited action (partially expanded), or a new Q node successor state is sampled that is not represented by another node of the same layer, the selection phase ends.

   A commonly used tree policy (**and the one we used**) that is synonymously used with MCTS is Upper Confidence Trees (UCT) (Kocsis & Szepesvári, 2006), which selects an action that maximizes the Upper Confidence Bound (UCB) value. Let $s \in S$ and $V_a, N_a$ with $a \in \mathbb{N}$ be the return sum and visits and of the Q nodes of the node representing $s$. The UCB value of any action $a$ is then given by

$$
\text{UCB}(a) = \underbrace{\frac{V_a}{N_a}}_{\text{Q term}} + \underbrace{\lambda \sqrt{\frac{\log\left(\sum_{a' \in \mathbb{A}(s)} N_{a'}\right)}{N_a}}}_{\text{Exploration term}} . \tag{19}
$$

   The exploration term quantifies how much the Q term could be improved if this Q node was fully exploited and is controlled by the exploration constant $\lambda \in \mathbb{R} \cup \{\infty\}$. If one chose $\lambda = 0$, the UCT selection policy becomes the greedy policy and for $\lambda = \infty$, the selection policy becomes a uniform policy over the visits. In case of equality, some tiebreak rule has to be selected, which is typically a random tiebreak. From here, will use MCTS and UCT (MCTS with UCB selection formula) synonymously.

2. **Expansion**: Unless the selection phases ended in a terminal state node, the search directed acyclic graph is expanded by a single node. In case the selection phase ended in a partially expanded state node, then one unexpanded action is selected (e.g. randomly, or according to some rule), the corresponding Q node is created and added as a child and one successor state of that Q node is sampled and added as a child to the new Q node. If the selection phase ended because a new successor of a Q node was sampled, then a state node representing this new state is added as a child to that Q node.

3. **Rollout/Simulation phase**: Starting at the state $s_{rollout}$ of the newly added state node of the expansion phase (or at a terminal state node reached by the selection phase), actions according to the *rollout policy* are repeatedly selected and applied to $s_{rollout}$ until a terminal state is reached. All states encountered during this phase are not added to the search graph.

4. **Backpropagation**: In this phase, the statistics of all Q nodes that were part of the last sampled trajectory that corresponds to a path in the search graph are updated by incrementing their visit count and adding the trajectory's return (of the trajectory starting at the respective Q node) to their return sum statistic.

## A.13 PROBLEM DESCRIPTIONS

We will provide a brief description of each of the IPPC problems used for the experiments. Note that most environments are parametrized; the concrete setups we used can be found in the *ExperimentConfigs* folder of our publicly available implementation (Authors, 2025).

- **Academic Advising**: The Academic Advising domain was used for the IPPC 2014 (Grzes et al., 2014). The agent is a student whose goal is to pass certain academic classes. Formally, the state is an element in $\{P, \text{NP}, \text{NT}\}^n$ (representing for each course whether it has been passed, not been passed, or not been taken) and the agent's action is to choose a course to take. The course outcome depends on the states of the prerequisite courses. The episode ends, when all courses are passed, and while not all mandatory courses, a subset of all courses, are passed, the agent incurs a constant penalty per step.

- **Connect4**: Connect 4 is played on grid with 7 columns and 6 rows. Each turn, one player places a stone of its color in one of the columns that is not yet filled with stones. The stone occupies the first cell in the chosen column that is not yet occupied.

- **CooperativeRecon**: This domain models a robot having to prove the existence of life on a foreign planet. The robot is modeled as moving on a 2-dimensional grid which contains a number of objects of interest and a base. If the agent is at an object of interest, it can survey the object for the existence of water and life. The probability of a positive result of the latter is dependent on whether water has been detected. If life has been detected, the agent may photograph the object of interest which is the only way to gain a reward. Each detector may break on usage making it either unusable or decreasing its chance of working. The detectors can be repaired at the base.

- **Constrictor**: Constrictor is played on an $n$ times $n$ grid. Players take turns moving to any of the neighboring (4-neighborhood) grid cells that neither moves the player out of bounds nor hits any cell that has already been visited by any of the two players. The game ends when one player has nowhere left to move.

- **Chess**: Chess is played on an 8-by-8 board, with each player beginning the game with an identical set of 16 pieces, each with distinct movement abilities. Players alternate turns, moving one piece per turn according to its specific movement rules. A piece can capture an opponent's piece by moving to the square it occupies, thereby removing it from the board. The game concludes when a player captures the opposing king, which is one of the pieces in play.

- **EarthObservation**: EarthObservation was a test problem for the IPPC 2018 which models a satellite orbiting earth. Formally, each state is a position on a 2-dimensional grid, representing the satellite's longitudinal position and the latitude the camera is aimed at as well as weather levels for some designated cells. At each step, the weather levels stochastically change independent of the agent's actions which are to idle, to take a photo of the current position, or increment/decrement the current cells $y$-position (i.e. shifting the camera focus). A reward is obtained if one of the designated cells is photographed with an amount depending on the cell's current weather condition.

- **Game of Life**: The original game of life by John Conway (Gardner, 1970) is a cellular automaton and modified into a stochastic MDP as a test problem for the International Probabilistic Planning Competition (Sanner & Yoon, 2011) by introducing noise to the deterministic state transition, setting the current number of alive cells as the reward, and allowing the agent to choose one cell which will contain a living cell with a high probability. States are elements in $\{0, 1\}^{n \times n}$ describing whether there is an alive cell at each cell on a grid. To reduce the action space that scales quadratically which the grid length, we allow only a subset of the original actions, which is to specify one alive cell that is prevented from dying.

- **Manufacturer**: In this domain, the agent manages a manufacturing company. The agent's ultimate goal is to sell goods to customers. However, to sell a good, the agent has to first produce the good, which may require building factories and acquiring the necessary goods required for production. Additional difficulty comes from the fact that the goods' price levels vary stochastically.

- **Navigation**: Navigation was a test problem for the International Probabilistic Planning Competition 2011 (Sanner & Yoon, 2011). The goal is to move a robot on an $n \times m$ grid from $(n, 1)$ to $(n, m)$ in the least number of steps. The robot may move to any of the four adjacent tiles, however, each tile is assigned a unique probability with which the robot is reset back to $(n, 1)$. At each step, except the one where the goal is reached, the agent incurs a constant negative reward, making the objective to reach the goal state as quickly as possible. *Note that the concrete instance we used is not the cherry-picked motivational example from Section 3 but rather an instance, already used in the literature (Anand et al., 2016).*

- **Numbers Race**: Numbers Race, players take turns choosing an integer between 1 and $n \in \mathbb{N}$. The goal is to choose a number $m \leq n$ such that the sum of all previous numbers is equal to some goal number $g \in \mathbb{N}$. If this sum exceeds $g$, then the player who overshot, loses. For this paper, we used $g = 200, m = 15$.

- **Othello**: Othello is usually played on an 8x8 (in our case, for simplicity's sake on a 6x6) board where players take turns placing a stone of their color on an empty cell. Once placed, all opponent's stones are flipped that are contained in a vertical, horizontal or diagonal line that starts at the placed stone and that ends at the first stone of the same color as the placed stone going in the line's direction. When neither player has an available move, the player with the most stones wins.

- **Pylos**: Pylos is played on an initially empty 4x4x4-dimensional grid on which players take turns placing stones of their own color. Each player starts with 15 stones, which can be placed at any empty cell that is either at the bottom or contains exactly 4 stones in the layer beneath it (i.e., the stone requires a foundation). Instead of placing one of one's remaining stones, one may also move a stone of one's color any number of layers upward as long as they are not part of the foundation for another stone. The game ends when one player runs out of stones or when one player completes the pyramid (bottom layer is 4x4, then 3x3, ..., 1x1).

- **Quarto**: Quarto is played on a 4x4 grid. The goal is to complete a vertical, horizontal, or diagonal line of length 4 with stones that all share a common property. There are initially 16 stones that can be placed on the board. The set of stones is given by $S = \{0, 1\}^4$ where the $i$-th component of a stone is referred to as a property. Players take turns placing a stone and then choosing one of the remaining, not yet placed stones which the opponent must place in the next turn (the game starts with one player selecting a stone for the opponent).

- **Racetrack**: Racetrack was first described by Martin Gardner (Gardner, 1973) where the goal is to move a car in as few steps as possible to a goal-position on a graph starting from a random position. Each state is a tuple in $\mathbb{Z}^2 \times \mathbb{Z}^2$ that represents the current position and velocity vector. The available actions are adding a vector from $\{-1, 0, 1\}^2$ to the velocity vector (i.e. accelerating). At each step, with some probability, the chosen acceleration vector is replaced by $(0, 0)$. The state transitions by first adding the acceleration vector to the current velocity vector and then adding the velocity vector to the position vector. If a boundary or an obstacle has been hit, the car's position is reset to one of the initial positions with zero velocity. Some implementations however, set the velocity vector to zero and place the car at an empty tile closest to the crash position that is on the straight line connecting the last position to the crash position (Anand et al., 2016).

- **Sailing Wind**: Originally proposed by Robert Vanderbei (Vanderbei, 1996), the goal of Sailing Wind is to move a ship that starts at $(1, 1)$ on an $n \times n$ grid to $(n, n)$ with minimal cost. There is no consistent use of a transition and reward function throughout the literature. There may just be two available actions (*down*, *right*) (Jiang et al., 2014) or up to seven (each adjacent cell except the one facing a stochastic wind direction) (Anand et al., 2015). The cost of each action is dependent on the current wind direction which stochastically changes its direction at each step independent of the player's actions.

- **SysAdmin**: Used as a test problem for the IPPC 2011, a SysAdmin instance is a graph (describing a network topology) with $n \in \mathbb{N}$ vertices. The state space is $\{0, 1\}^n$ (describing which machines are currently operating) and the action space is $\{1, \ldots, n\}$ (describing with machine to reboot). At each step, the reward is dependent on the machines that are currently working, a reboot causes the rebooted machine to have a high chance of working in the next step. Machines can randomly fail at each step, however this probability is increased when a neighbor fails.

- **Tamarisk**: Tamarisk is yet another problem from the IPPC 2014 (Grzes et al., 2014) which models the expansion of an invasive plant in a river system. The river system is modelled as a chain of reaches where each reach contains a number of slots that may be unoccupied, occupied by a native plant, or occupied by the invasive Tamarisk plant. Both plant types spread stochastically to neighboring states with a higher probability of spreading downstream. At each time step, the agent chooses an action for one reach, which are doing nothing, eradicating Tamarisk, or restoring a native plant. The action chosen at a reach is applied to all slots in that reach. Except for the do-nothing action, all actions can randomly fail. The agent has to balance the action's costs with the penalties incurred for existing Tamarisk plants.

- **Triangle Tireworld:** Tireworld was proposed as a test problem for the IPPC 2004 (Younes et al., 2005). In the original goal-based version, the agent is a car that traverses a graph. At

each step, the car may move to an adjacent node, change its tire, or load a tire. The goal is to reach a designated goal node. At each step, the car's tire may randomly break. If the car isn't carrying a spare tire, the goal can no longer be reached. Otherwise, if available, a spare tire (at most one can be carried) must be used to replace the current tire. Some nodes contain spare tires, which when the agent visits them, can be picked up.

- **Skills Teaching**: This domain models a student-teacher interaction, where the agent plays the role of the teacher. There is a fixed number of skills that form a directed graph of prerequisites. The student possesses one of three levels of sufficiency at each skill. The agent is rewarded for each skill being at the highest sufficiency and punished for each skill at the lowest sufficiency level. At each, step the agent may choose a skill for which to pose a question to the student or give the student a direct hint. The student can increase their sufficiency at that skill for correctly answering a question and lose sufficiency for answering wrong. The probability of getting a question right is dependent on the sufficiency of the skill's prerequisite. A hint can elevate the student to the medium sufficiency level directly but only if all prerequisites are at the highest sufficiency.

- **Traffic**: This problem models a traffic system in which the agent is tasked with controlling/advancing intersections with the goal of minimizing congestion. The traffic system is modeled as a directed graph and each vertex is either empty or occupied. Occupancy flows along the graph's edges except for some designated intersection edges where the flow is dependent on the intersection's state. The only stochasticity of this MDP arises in the form of cars spawning randomly at the designated perimeter vertices. The agent receives a reward equal to the negative number of occupied vertices that have one predecessor vertex that is also occupied.

