# OpenReview forum: "Discovering state equivalences in UCT search trees by action pruning"
_ICLR.cc/2026/Conference — ICLR 2026 Conference Withdrawn Submission_

### Official Review · Reviewer_7qcc · 2025-10-30

**Soundness:** 2
**Presentation:** 1
**Contribution:** 1
**Rating:** 0
**Confidence:** 4

**Summary:**

This paper is tries to improve on abstractions used for reducing the search space in Monte Carlo tree search (MCTS). Typically, in large, sequential decision spaces MCTS employs abstraction methods to aggregate states to reduce the search space. However, the authors claim that little work is done in stochastic settings or settings with large action spaces.  They show how one such abstraction technique, On the Go Abstractions in Upper Confidence bounds applied to trees (OGA-UCT) fails in these settings. The authors then provide a remedy to this problem by introducing IPA-UCT that addresses a key issue that causes OGA-UCT to fail. The authors then provide experimental results in a number of domains to show how their approach improves on OGA-UCT.

**Strengths:**

- The evaluation design was done well. The authors ran their experiments on a wide array of problem domains and with a suitably high number of random seeds to demonstrate statistical significance.
- The motivation for the paper is strong. State abstraction remains a challenging problem in planning and methods to do so are of great interest to the community.
- Although, I only skimmed the proof, the claim of why ASAP finds few abstractions seems to supported by their theory.

**Weaknesses:**

- The main weakness with the paper is its lack of significance. The method the authors propose, pruning actions with UCB values less than $Q_{max}$, seems to effectively boil down to considering only the $k$ actions with the highest UCB values for search and abstraction for some arbitrary value of $k$. This minor change is not enough to be considered a significant contribution.
- If we consider the argument they show for why ASAP fails, the authors do not clearly explain how IPA-UCT explicitly overcomes this nor do they link it back to the theory (Eq (7)) they provide.
- Another reason, I believe the contributions are not clear is because they discuss their contributions only briefly. The authors spend almost 3 pages describing the background (Sec 2). The space could be better used to focus on their contributions.
- Additionally, the background section was a tough read. It was difficult to understand. I did not quite understand the intent behind how they structured the section but there was no flow in information. They do not formally define equivalence, or equivalence classes. The definition of equivalence relations is really confusing as it defined double-recursively and the authors do not provide any base cases (see Eqs 1, 2 and 3).
- I also do not see how what they describe is different from the concept of bisimilarity. If it is the same, why do they not base their description using that formalism?
- The example using the Navigation problem domain does not seem to be correct. Firstly, while the optimal path is to follow 3->8->13->18, this path does not yield an average return of -3. It would do so if cell 8 had a 0.0 probablity of a reset. However, with a non-zero reset chance, the average return will be < -3. Secondly, according to the definition of ASAP-equivalence in Eq (6), the two suboptimal paths would indeed be abstracted. The definition states that as long as each state has at least one action leading to an identical outcome, they
would be abstracted. For this example, 12->13 would be equivalent to 14->13 according to Eq (6). Am I missing or misunderstanding something?
- In the experiments, Figs 3 and 4 show plots for three algorithms. However, the description discusses four algorithms. Is one missing or are two plots joined together?
- Maybe I missed it, but I could not find an explanation of what pairing values are.
- In figures 5 and 6, the authors state there is a downward trend with the value of $\lambda_p$. Given what the plots look like, I do not think that is a statement that can confidently be made. If we were to make that statement anyway, it would only apply to pairing values. The improvement value plots seem to get flat with an increase in the number of iterations. It seems the pruning factor makes no difference if we give search enough samples, which makes sense. Overall, the plots are very erratic and I don't know what to make of the results described.

**Questions:**

- You state that you plot the 99% confidence intervals. I do not see any error bars or anything. Is it because the intervals are too small at the scale at which the figures are plotted?
- What is a pairing value?
- What is an abstract Q node?
- Why are you stating that $s$ is a successor to $s'$ when conventionally it goes the other way? This adds to the unreadability of Sec 2.
- "Later, when the different OGA variants are experimentally investigated, one ablation that will also be conducted is to test the performance of random state abstractions to ensure that any performance gains due to the usage of abstractions are better than if random abstractions were used." Where do you do this? Is this in Figs 3 and 4? Are they improvements over random abstractions or no abstractions?

---

### Official Review · Reviewer_mCkK · 2025-11-01

**Soundness:** 3
**Presentation:** 2
**Contribution:** 3
**Rating:** 6
**Confidence:** 4

**Summary:**

Authors propose IPA, a novel MCTS abstraction mechanism to foster more state abstraction than in the ASAP baseline. While ASAP abstracts state nodes when all their actions are equivalent, IPA abstracts state nodes just when their expected optimal actions are equivalent. IPA is merged with OGA-UCT into IPA-UCT, which constantly outnumbers the states OGA-UCT abstracted and constantly outperforms OGA-UCT for various iteration budgets on diverse board games. Authors also propose conceptual categorization of abstraction methods, positioning IPA and ASAP branched from p(runed)-ASAP whose parent is ASASAP.

**Strengths:**

The problem statement is clearly described. The proposed method is simple and directly complement the baseline (ASAP) and is theoretically well established. Empirical results show constant improvements compared to the baseline. Limitations of IPA are well discussed.

**Weaknesses:**

### Major Weaknesses

- Writing and Readability Issues: The main idea of this paper is novel and sound, but the most degrading part is the writing. Once understood, the theoretical and empirical results are convincing, but reaching that level of understanding required several careful readings. Some sentences are overly lengthy or colloquial, making the paper unnecessarily exhausting to read.

- Delayed Introduction of the Main Method (IPA): The exact mechanism of the proposed method, IPA, does not appear until page 6. While several preliminary explanations are understandable, the excessive focus on foundations and previous methods distracts from the main contribution. Since the main suggestion of the paper is IPA, providing a brief abstract explanation of the IPA mechanism earlier (e.g., in the Introduction) would improve readability and focus.

- Structural and Organizational Issues: The new formulations of ASASAP and p-ASAP on pages 2–3 are difficult to follow due to repetitive and unfamiliar terminology. These parts may be theoretically useful, but positioning them later in the Method or Discussion section would improve clarity. Section 2 also includes paragraphs (“OGA-UCT for multi-agent settings,” “OGA-UCT extensions to high stochasticity,” and “RSTATE-OGA”) that seem more appropriate for the Experimental Setup. In the Method section, describing IPA first and then comparing it to ASAP would create a smoother and more logical flow.

### Minor Weaknesses
- Although it is not necessarily a weakness, I suggest adding some actual state abstraction examples that ASAP could not detect but IPA did, based on the environments used in experiments. While a conceptual example is already described on page 6 with the grid navigation task, presenting actual examples would more effectively support the proposed claims. Additionally, showing faulty abstraction examples from IPA would help illustrate its potential limitations.
- Line 277, Self-reference error (“Section 3”)

**Questions:**

- What is the meaning of the sentence in line 147 “Hence, any abstraction algorithms using these, need to be slightly modified”?
- What could be the definition of the words “correct abstractions” in line 149? (feels a bit ambiguous)
- Did IPA handle stochastic environments well?

---

### Official Review · Reviewer_aW4M · 2025-11-09

**Soundness:** 3
**Presentation:** 2
**Contribution:** 2
**Rating:** 4
**Confidence:** 3

**Summary:**

The paper identifies a problem with ASAP/OGA-UCT: they rarely find state abstractions because they require matching *all* actions between states. The authors propose IPA (Ideal Pruning Abstractions), which only compares near-optimal actions, and implement this as IPA-UCT by using a UCB-based pruning set within OGA-UCT. The theoretical motivation is a combinatorial bound (Equation 7) showing ASAP's strictness, illustrated with a Navigation example. Experiments across 20+ domains show consistent but modest improvements in aggregate metrics (pairings and relative improvement), with 5-9% runtime overhead. Soundness is proven asymptotically.

**Strengths:**

- **Clear problem diagnosis**: The combinatorial bound (Equation 7) and toy example effectively demonstrate why OGA finds almost no state abstractions. Table 2 in the appendix backs this up empirically.

- **Simple, practical solution**: The UCB-based pruning (Equation 10) is easy to implement and aligns naturally with how UCT allocates samples.

- **Thorough empirical evaluation**: Testing on many domains with multiple budgets, proper confidence intervals (99%), and generalization-focused metrics (pairings/relative improvement) rather than just per-environment tuning.

- **Low overhead**: The ~5-9% runtime cost (Table 1) is reasonable for the gains achieved.

**Weaknesses:**

**Major issues:**

1. **Missing finite-sample analysis**: The soundness guarantee is only asymptotic. There's no analysis of how often J_UCB prunes optimal actions at realistic visit counts (100-1000). This is critical since the method specifically targets low-sample regimes.

2. **Modest empirical gains**: While aggregate improvements are consistent, per-environment parameter-optimized results (Figures 6-7, Appendix A.2) often show ties or minimal improvements. The authors acknowledge gains are small even in the best cases. For ICLR, I'd expect either stronger empirical results or more substantial theory.

3. **Parameter sensitivity without adaptive solution**: Performance depends heavily on λ_p (Figure 5), but there's no adaptive scheduling mechanism—just grid search. The paper notes this as a limitation but doesn't attempt simple solutions like annealing λ_p with depth or visit counts.

**Minor issues:**

4. **Equivalence relation approximation**: J_UCB doesn't induce an equivalence relation, requiring the heuristic representative/tie-break mechanism (Appendix A.11). This is central to making the method work but relegated to the appendix.

5. **Limited positioning**: Related work mentions refinement-based abstractions (PARSS) and Elastic-MCTS but doesn't compare against them or clearly delineate where IPA fits in the landscape of approximate abstraction methods.

**Questions:**

### Questions for Authors

1. Can you provide empirical pruning error rates? How often does J_UCB prune an actually optimal action at typical visit counts (100, 500, 1000) across several domains? Even without a formal bound, this data would help assess practical risk.

2. How stable are state clusters under your equivalence closure heuristic (A.11)? Can you report "cluster churn"—how often states switch abstract nodes as search progresses?

3. Why not compare against PARSS or Elastic-MCTS on a few domains? This would help situate IPA's contribution relative to other recent abstraction methods.

4. The Navigation example is compelling, but it seems somewhat cherry-picked. Can you identify which environments from your test suite have similar structure where IPA should theoretically help, and which don't, then show this correlates with your empirical results?

### Suggestions for Improvement

- **Add finite-time analysis**: Even a simplified analysis would substantially strengthen the paper.

- **Move A.11 to main text**: The equivalence closure heuristic is essential to understanding what IPA-UCT actually does. Summarize it in the main paper with discussion of trade-offs.

- **Clarify positioning**: Add a paragraph explicitly connecting to distance-based and bisimulation-style abstractions as complementary approaches that also relax strict equivalence. You don't need new experiments—just help readers understand where IPA fits.

- **Method section**: Can be restructured or writing can be improved to remove weird phrase switches like "theory:" / "empirical results".

### Recommendation

The core idea is sensible and the experimental work is careful. However, the paper sits uncomfortably between theory and practice: the theory is limited to asymptotic guarantees that don't address the finite-sample regime where the method operates, and the empirical gains over strong OGA baselines are modest when parameters are tuned per-environment.

I could move toward acceptance if the authors provide: (1) finite-sample analysis or at least empirical pruning error rates, (2) evidence that adaptive λ_p can achieve robust performance, or (3) clearer positioning and comparison against other recent abstraction methods. The current contribution feels incremental.

---

### Official Review · Reviewer_6B9E · 2025-11-16

**Soundness:** 2
**Presentation:** 2
**Contribution:** 2
**Rating:** 6
**Confidence:** 3

**Summary:**

The paper proposes a framework for relaxed state abstraction that can be flexibly applied to noisy or large action space. The authors propose Ideal Pruning Abstractions in UCT (IPA-UCT) that prunes suboptimal actions to reveal additional state equivalences. They formalize the underlying theoretical framework by introducing IPA, p-ASAP, and ASASAP abstractions, generalizing existing abstraction schemes. Empirically, IPA-UCT consistently improves sample efficiency and planning performance across diverse decision-making domains.

**Strengths:**

- The motivation of the proposed framework is intuitive and understandable. The paper also provides a principled extension of the abstraction hierarchy which is reasonable.
- The authors demonstrate why prior methods struggle to discover meaningful state abstractions and how their framework mitigates this limitation.
- Experiments show the effectiveness of the proposed method. The authors also include ablations on a few hyperparameters. Finally, the paper is well written and easy to follow.

**Weaknesses:**

- I think it is good to include more intuition or visualization on how the discovered abstractions influence search behavior in more complex domains.
- It is questionable that the proposed framework could be extended to more complex high-dimensional environments. Could the authors include discussions on this?

**Questions:**

1. Could the authors provide more concrete examples or visualizations showing how IPA-UCT’s discovered state abstractions differ qualitatively from those of OGA-UCT?
2. Is there a principled or heuristic way to learn the hyperparameters (e.g., lambda) adaptively during search to improve generalization across domains?
3. Could the authors include standard deviations in the main experiments?

---

### Note · Authors · 2025-11-18

I have read and agree with the venue's withdrawal policy on behalf of myself and my co-authors.